# Metformin Is a Pyridoxal-5′-phosphate (PLP)-Competitive Inhibitor of SHMT2

**DOI:** 10.3390/cancers13164009

**Published:** 2021-08-09

**Authors:** Angela Tramonti, Elisabet Cuyàs, José Antonio Encinar, Matthias Pietzke, Alessio Paone, Sara Verdura, Aina Arbusà, Begoña Martin-Castillo, Giorgio Giardina, Jorge Joven, Alexei Vazquez, Roberto Contestabile, Francesca Cutruzzolà, Javier A. Menendez

**Affiliations:** 1Institute of Molecular Biology and Pathology, National Research Council, 00185 Rome, Italy; angela.tramonti@cnr.it; 2Department of Biochemical Sciences, Sapienza University of Rome, 00185 Rome, Italy; alessio.paone@uniroma1.it (A.P.); giorgio.giardina@uniroma1.it (G.G.); roberto.contestabile@uniroma1.it (R.C.); 3Girona Biomedical Research Institute (IDIBGI), 17190 Girona, Spain; ecuyas@idibgi.org (E.C.); sverdura@idibgi.org (S.V.); aarbusa@idibgi.org (A.A.); bmartin@iconcologia.net (B.M.-C.); 4ProCURE (Program Against Cancer Therapeutic Resistance), Metabolism & Cancer Group, Catalan Institute of Oncology, 17007 Girona, Spain; 5Institute of Research, Development and Innovation in Biotechnology of Elche (IDiBE) and Molecular and Cell Biology Institute (IBMC), Miguel Hernández University (UMH), 03202 Elche, Alicante, Spain; jant.encinar@umh.es; 6Max Plank Institute for Molecular Genetics, 14195 Berlin, Germany; matthias.pietzke@mdc-berlin.de; 7Unit of Clinical Research, Catalan Institute of Oncology, 17007 Girona, Spain; 8Department of Medicine and Surgery, Universitat Rovira i Virgili, 43201 Reus, Spain; jorge.joven@salutsantjoan.cat; 9Unitat de Recerca Biomèdica (URB-CRB), Hospital Universitari de Sant Joan, Institut d’Investigació Sanitaria Pere Virgili, Universitat Rovira i Virgili, 43201 Reus, Spain; 10The Campus of International Excellence Southern Catalonia, 43003 Tarragona, Spain; 11Cancer Research UK Beatson Institute, Glasgow G61 1BD, UK; avazque1@protonmail.com

**Keywords:** one-carbon metabolism, serine, glycine, folate, serine hydroxymethyltransferase

## Abstract

**Simple Summary:**

The mitochondrial enzyme serine hydroxymethyltransferase (SHMT2), which converts serine into glycine and generates 1C units for cell growth, is one of the most consistently overexpressed metabolic enzymes in cancer. Here, we reveal that the anti-diabetic biguanide metformin operates as a novel class of non-catalytic SHMT2 inhibitor that disrupts the pyridoxal-5′-phosphate (PLP)-dependent SHMT2 oligomerization process and ultimately SHMT2 activity. As SHMT2 inhibitors have not yet reached the clinic, these findings may aid the rational design of PLP-competitive SHMT2 inhibitors based on the biguanide skeleton of metformin.

**Abstract:**

The anticancer actions of the biguanide metformin involve the functioning of the serine/glycine one-carbon metabolic network. We report that metformin directly and specifically targets the enzymatic activity of mitochondrial serine hydroxymethyltransferase (SHMT2). In vitro competitive binding assays with human recombinant SHMT1 and SHMT2 isoforms revealed that metformin preferentially inhibits SHMT2 activity by a non-catalytic mechanism. Computational docking coupled with molecular dynamics simulation predicted that metformin could occupy the cofactor pyridoxal-5′-phosphate (PLP) cavity and destabilize the formation of catalytically active SHMT2 oligomers. Differential scanning fluorimetry-based biophysical screening confirmed that metformin diminishes the capacity of PLP to promote the conversion of SHMT2 from an inactive, open state to a highly ordered, catalytically competent closed state. CRISPR/Cas9-based disruption of SHMT2, but not of SHMT1, prevented metformin from inhibiting total SHMT activity in cancer cell lines. Isotope tracing studies in SHMT1 knock-out cells confirmed that metformin decreased the SHMT2-channeled serine-to-formate flux and restricted the formate utilization in thymidylate synthesis upon overexpression of the metformin-unresponsive yeast equivalent of mitochondrial complex I (mCI). While maintaining its capacity to inhibit mitochondrial oxidative phosphorylation, metformin lost its cytotoxic and antiproliferative activity in SHMT2-null cancer cells unable to produce energy-rich NADH or FADH_2_ molecules from tricarboxylic acid cycle (TCA) metabolites. As currently available SHMT2 inhibitors have not yet reached the clinic, our current data establishing the structural and mechanistic bases of metformin as a small-molecule, PLP-competitive inhibitor of the SHMT2 activating oligomerization should benefit future discovery of biguanide skeleton-based novel SHMT2 inhibitors in cancer prevention and treatment.

## 1. Introduction

A close relationship exists between the anti-cancer activity of the biguanide metformin and the folate-dependent functioning of the serine/glycine one-carbon (SGOC) network [1,2,3]. The elevation of homocysteine levels, an archetypal marker of one-carbon (1C) deficiency due to impaired flux through the serine catabolism pathway [4], is a pharmacodynamic marker that informs tumor response to metformin in non-diabetic patients with breast cancer [5]. Metformin treatment phenocopies the metabolic effects of hindering the incorporation of 1C units into folates by serine hydroxymethyltransferases (SHMTs). Knockdown of SHMT2, the mitochondrial SHMT isoenzyme that donates a 1C unit from serine to bioactive tetrahydrofolate (THF), triggers the accumulation of precursors upstream of inosine-monophosphate (IMP) prior to incorporation with the THF-bound 1C unit [6,7,8]. Treatment of cancer cells with metformin similarly promotes a build-up of IMP biosynthetic intermediates such as formimino-THF [9], suggesting the impairment of its further metabolism by SHMT2. Moreover, SHMT2-driven mitochondrial serine metabolism is an indispensable facet of the ability of metformin to promote S-adenosylmethionine (SAM) accumulation and global DNA methylation [10]. Intriguingly, metformin treatment leads to complex responses that can improve or impede the anti-tumor effects of serine and glycine starvation in vitro and in vivo [11,12].

Most of the aforementioned effects of metformin on the channeling of folate-bound 1C pools through the mitochondrial catabolism of serine are commonly viewed as indirect, downstream effects of its capacity to target mitochondrial complex I (mCI) [13,14,15,16]. Given that metformin does not affect the de novo synthesis of serine, the metabolic synthetic lethality between metformin and serine starvation in cultured cancer cells has been explained in terms of the requirement for serine to allow cells to compensate for the metformin-induced reduction in oxidative phosphorylation (OXPHOS) [11]. In vivo, serine availability might also influence the antiproliferative effects of metformin via an increased dependency on (metformin-responsive) mitochondrial OXPHOS that becomes necessary to maintain ATP production when de novo serine synthesis is activated in response to serine starvation [12]. Disruption of mitochondrial respiratory chain function by mCI blockade has also been proposed as the sole mechanism underlying the ability of metformin to target the mitochondrial catabolic release of serine-derived 1C units such as formate [17]. An alternative explanation, which we explore here, is that the anti-folate mimicry activity of metformin might involve the direct targeting of specific SGOC enzymes in cancer cells.

Using a systematic approach combining radiometric enzymatic assays, in silico computational biology, differential scanning fluorimetry-based biophysical screenings, and CRISPR/Cas9-based SHMT gene dropout studies, and isotope tracing, we now report that metformin, at millimolar concentrations, has a novel function as a non-catalytic pyridoxal-5′-phosphate (PLP)-competitive inhibitor of mitochondrial SHMT2. This quantitative characterization of the interaction between metformin and SHMT2 may aid the rational design of more effective SHMT2 inhibitors originating from the same biguanide skeleton.

## 2. Materials and Methods

### 2.1. Reagents

Metformin (1,1-dimethylbiguanide hydrochloride) was purchased from Sigma-Aldrich (Cat. #D150959, St. Louis, MO, USA) and dissolved in sterile water to make a 1 mol/L stock solution. Formyl-tetrahydrofolate was kindly provided by Merck and Cie (Schaffhausen, Switzerland).

### 2.2. Enzyme Purification

Recombinant human SHMT1 and SHMT2 were purified as described [18]. Apo-proteins were prepared using L-cysteine as described [19].

### 2.3. Inhibition Experiments with Purified Recombinant SHMT Isoforms

Inhibition of SHMT activity by metformin was assessed using an established competitive binding assay [20,21]. The assay is based on the spectrophotometric measurement of the quinonoid intermediate that develops when both glycine and formyl-THF bind to SHMT, forming an enzyme/glycine/formyl-folate ternary complex [20]. The quinonoid intermediate, which has an intense absorption band with a maximum at ~500 nm, derives from the deprotonation of glycine, but it can accumulate to a measurable extent solely when a folate ligand is also bound to SHMT and a ternary complex is formed [22]. Accordingly, the absorbance at 500 nm is proportional to the fraction of enzyme present as a ternary complex. Metformin was dissolved in water to 50 mg/mL (0.3 mol/L).

Exploratory experiments were performed with 5 μmol/L solutions of purified human recombinant SHMT1 and SHMT2, which were incubated with a saturating concentration of glycine (3 mmol/L) in the presence of 5 μmol/L formyl-THF and a low concentration range of metformin (0–30 mmol/L) in 20 mmol/L potassium phosphate buffer, pH 7.2. All measurements were performed in triplicate. Inhibition curves were fitted to Equation (1), in which [*I*] is the concentration of metformin, to obtain the observed apparent inhibition constants (*K_i_*).
(1)$$\%\;\mathrm{Activity}=100\times \left(1-\left[I\right]/\left(\left[I\right]+K\_i\;\right)\right)$$

In more complete assays with SHMT2 (5 μmol/L) enzyme solutions, when formyl-THF was the varying ligand (2.5–160 μmol/L), glycine was maintained at 10 mmol/L, and when glycine was the varying ligand (0.04–10 mmol/L) the formyl-THF concentration was maintained at 160 μmol/L. In both cases, the buffer used was 20 mmol/L potassium phosphate buffer (pH 7.8), and metformin concentrations varied between 0 and 200 mmol/L. In all assays, formyl-THF was added as the last component and, after rapid manual mixing, the absorbance change at 502 nm was recorded using a Hewlett-Packard 8453 diode-array spectrophotometer (Agilent Technologies, Santa Clara, CA, USA). 

The data obtained from the two series of experiments were employed to generate saturation curves and double reciprocal plots, which were globally fitted to Equation (2).
(2)$$1/v\_0=\left(\alpha K\_M\right)/V\_MAX\;1/\left[S\right]+\alpha ’/V\_MAX\;$$
in which *v_o_* is the initial velocity, K_M_ is the Michaelis–Menten constant, V_MAX_ is the maximum velocity and
$$\alpha =1+\left[I\right]/K\_i\;$$
$$\alpha \_1=1+\left[I\right]/K\_i1$$

### 2.4. Differential Scanning Fluorimetry (DSF)

Thermal denaturation assays were performed using 0.5 μmol/L SHMT apo-enzyme samples incubated with or without 200 mmol/L metformin in the presence of Sypro™ Orange (5×, Thermo Scientific). SHMT1 and SHMT2 holo-enzyme samples were obtained by addition of 3 and 10 μmol/L PLP to apo-enzyme samples, respectively. DSF assays were performed on Real Time PCR Instrument (CFX Connect Real Time PCR system, Bio-Rad). Fluorescence was measured from 25 to 95 °C in 0.4 °C/30 s steps (excitation 450–490 nm; detection 560–580 nm). All samples were run in triplicate. Denaturation profiles were analyzed following removal of those points representing quenching of the fluorescent signal due to post-peak aggregation of protein–dye complexes. All curves were normalized and fitted to a sigmoidal equation (Equation (3)) to obtain the melting temperatures (*T_m_*).
(3)Fluorescence=F_1 F_2−F_1/1+e^T_m−X/S 

Alternatively, the first derivative of the fluorescence emission as a function of temperature (−dF/dT) was plotted by using the Bio-Rad CFX manager software (Hercules, CA, USA).

For PLP titration experiments, PLP (0–10 μmol/L) was added to SHMT apo-enzymes samples (0.5 μmol/L) at 20 °C in 20 mmol/L potassium phosphate, pH 7.2 in the presence of Sypro™ Orange (5X Thermo Scientific) and with or without metformin (200 mmol/L). Melting temperatures, obtained from fitting of fluorescence of data to the Equation (3), were analyzed as a function of PLP concentration according to a sigmoidal equation.

### 2.5. Inhibition Experiments in Cells

Measurements of cellular SHMT activity were performed using a radioisotope assay based on the ability of SHMT to catalyze the exchange of the pro-2S proton of glycine with solvent [23]. Cells were detached using trypsin, centrifuged, and washed twice in 2 mL of PBS to remove growth medium. Then, 100 μL aliquots of cell suspensions were incubated with increasing concentrations of metformin (25–150 mmol/L) at 30°C for 1.5 h. [2-^3^H]glycine (23 μmol/L) was then added to samples, which were incubated for a further 4 h at 37 °C. Samples were then centrifuged to remove cells and reactions were stopped by the addition of 3% (*w*/*v*) trichloroacetic acid to remove radiolabeled glycine and radioactivity in the solvent was measured, as described [23]. Control reactions without cells were run in parallel to correct for background exchange as well as to measure 100% activity in those samples without added metformin. All measurements were performed in duplicate and normalized to cell numbers.

### 2.6. Molecular Docking Simulations

Crystallographic structures of human holo-SHMT1 (UniProt code: P34896, PDB code: 1BJ4) and human holo-SHMT2 (UniProt code: P34897, PDB code: 5V7I) were obtained from the Research Collaboratory for Structural Bioinformatics (RCSB) Protein Data Bank (PDB). FoldX 5.0 was used to determine the amino acids involved in the dimer-to-dimer interaction on structures 6FL5 (SHMT1) and 4PVF (SHMT2). The molecular structure of metformin was obtained from PubChem (ID: 4091). The specific edition of protein structures including water removal, PLP or mutated LLP (PLP L-peptide linking) to Lys, was made with PyMol software (PyMOL Molecular Graphics System, v2.3.3 Schrödinger, LLC, at http://www.pymol.org/ accessed on: 1 September 2019) without further optimization.

Molecular docking experiments were carried out using YASARA v19.9.17 software (Vienna, Austria) executing the AutoDock 4 algorithm with AMBER99 as a force field [24,25,26,27]. Briefly, a total of 50 flexible docking runs were set and clustered (7 Å) over the interfacial dimer-to-dimer surface and around the PLP binding cavity. The YASARA pH command was set to 7.4. The YASARA software calculates the Gibbs free energy variation (ΔG, kcal/mol), with more positive energy values indicating stronger binding.

### 2.7. Molecular Dynamics Simulations

YASARA dynamics v19.9.17 (Vienna, Austria) was employed to carry out all the MD simulations with AMBER14 as a force field. All simulation steps were run using a pre-installed macro (md_run.mcr) within the YASARA suite. Data were collected every 100 ps during 100 ns. The MM/PBSA calculations of solvation binding energy were calculated using the YASARA macro md_analyzebindenergy.mcr, with more negative values indicating instability. More details about the molecular docking procedures and MD simulations are available in previous works from our own group [28,29,30].

### 2.8. Cell Lines

HAP1 and HAP1shmt2KO cells (Cat. #HZGHC001954c006) were obtained from Horizon Discovery Ltd. (Cambridge, UK) and maintained at 37°C with 5% CO_2_ in IMDM medium (Gibco) supplemented with 10% fetal bovine serum (FBS), 2 mmol/L L-glutamine, and 100 IU/mL penicillin/streptomycin. HAP1shmt1KO cells were obtained from Dr. Ketan Patel [31] and were similarly cultured in IMDM medium supplemented with 10% FBS and 100 IU/mL penicillin/streptomycin.

### 2.9. Generation of NDI1- and SHMT2-Overexpressing Cell Lines

Overexpression of NDI1 in HAP1 and HAP1shmt1KO cells and
reintroduction of SHMT2 into the SHMT2-null HAP1 cell line were carried out by retroviral transduction using the PMXs-NDI1 and SHMT2_RES retroviral constructs (Addgene plasmids #72876 and #72880, respectively). Briefly, cDNA vectors were transfected into Plat-A retroviral packaging cells using FuGENE Transfection Reagent (Promega) and 9 μg of each plasmid. Media was changed 24 h after transfection and the virus-containing supernatant was collected 48 h after transfection. Viruses were passed through a 0.45 μm filter and stored at −80°C or used immediately. Target cells were infected in media containing 8 μg/mL polybrene. At 48 h post-infection, viruses were removed and cells were selected with 10 μg/mL blasticidin. Cells were grown for 2–3 weeks, and the resultant colonies were expanded and screened for gain of the relevant protein by immunoblotting.

### 2.10. Immunoblotting 

Cells were rinsed once in ice-cold PBS and harvested in a lysis buffer containing 150 mmol/L NaCl, 50 mmol/L Tris-HCl pH 7.4, 1 mmol/L EDTA, 1% Triton-X 100, 1 mmol/L phenylmethylsulfonyl fluoride, 1 mmol/L Na_3_VO_4_. Samples were sonicated for 1 min (under ice water bath conditions) with 2 s sonication and 2 s intervals to fully lyse cells and reduce viscosity. Protein content was determined by the Bradford protein assay (Bio-Rad, Hercules, CA, USA). Equal amounts of cellular protein were electrophoresed on 12% SDS-PAGE gels, transferred to nitrocellulose membranes, and incubated with an antibody against SHMT2 (Cat. # 12762S, Cell Signaling Technology, Inc., Danvers, MA, USA), followed by incubation with a horseradish peroxidase-conjugated secondary antibody, and chemiluminescence detection. β-actin (Cat. #66009-1-Ig, Clone #:2D4H5; Proteintech Group, Inc., Rosemont, IL, USA), was employed as control for protein loading.

### 2.11. Stable Isotope Labeling

Stable isotope experiments with HAP1 cells were performed in MEM, with the addition of 10% dialyzed FBS and all the amino acids and vitamins needed to mimic regular IMDM culture medium, including 400 μmol/L glycine, 2 μmol/L glutamine, 17 mmol/L glucose, and 400 μmol/L serine. To allow for adaptation to a different cell culture medium, cells were plated in triplicate in 12-well plates (150,000 cells/well) and cultured overnight before the start of experiments. Additional triplicate wells were seeded to count cells at the start and end of the experiment for normalization. The next day, the medium was removed, cells were washed once with PBS and then cultured for an additional 24 h in 1 mL of labeling medium containing either 400 μmol/L U-[^13^C]-Serine or 400 μmol/L [3-^13^C]-L-Serine (Cat. #CLM-1574-H and CLM-1572-01, respectively; Cambridge Isotope Laboratories, Inc., Tewksbury, MA, USA) in the absence or presence of graded concentrations of metformin. Media of the extracted cells were collected and centrifuged for 10 min at 13,000 g/4 °C to eliminate cellular debris; supernatants were transferred to new tubes and stored at −80 °C for later formate quantification. To determine consumption rates, labeling media were added to empty wells and incubated for the same time period. Formate extraction, derivatization and quantification was performed as described [32].

For the determination of intracellular metabolites, cells were washed with ice-cold PBS, extracted with acetonitrile/H_2_O_MQ_/methanol (ratio, 3:2:5) (liquid chromatography–MS (LC-MS) grade solvents), separated on a ZIC-pHILIC column (SeQuant, Merck KGaA) and detected using a Q Exactive Orbitrap mass-spectrometer (Thermo Fisher Scientific). The peak areas of different metabolites were determined using Thermo TraceFinder software, where metabolites were identified by the exact mass of the singly charged ion and by known retention times on the HPLC column against an in-house compound database created with commercial standards [32].

### 2.12. Cell Viability Assays

Cell viability effects of metformin were determined using the colorimetric MTT (3-4,5-dimethylthiazol-2-yl-2,5-diphenyl-tetrazolium bromide) reduction assay. Dose–response curves were plotted as a percentage of the control cell absorbance, which was obtained from control cells containing the vehicle (water) processed simultaneously. For each treatment, cell viability was evaluated as a percentage using the following equation: (OD_570_ of the treated sample/OD_570_ of the untreated sample) × 100. Sensitivity to agents was expressed in terms of the concentrations required for a 50% (IC_50_) reduction in cell viability. Since the percentage of control absorbance was considered to be the surviving fraction of cells, the IC_50_ values were defined as the concentration of drug that produced 50% reduction in control absorbance (by interpolation).

### 2.13. Real-Time Cell Growth

Proliferation effects of metformin were measured using the xCELLigence Real Time Cell Analysis (RTCA) DP instrument (ACEA Biosciences, San Diego, CA, USA). Briefly, cells were plated at 20,000 cells/well in 100 μL of fresh medium in an E-plate 16 (Cat. #300600890, Agilent). Initial attachment and growth were continuously monitored for approximately 24 h at 37 °C and 5% CO_2_ for stabilization. Then, 100 μL of medium was removed from each well and replaced with fresh medium with or without graded concentrations of metformin. The plate remained in the RTCA Station for 96 h and impedance was monitored every 5 min for approximately 24 h at 37 °C and every 15 min for the next 72 h. Growth curves were plotted using the RTCA Software Package 2.0 (xCELLigence RTCA, Roche Applied Science, Basel, Switzerland) and normalized to the time point of initial treatment; time-dependent cell index (CI), doubling time, and slope graphs were generated as per the manufacturer’s instructions. Three biological replicates were evaluated in each experiment, which permits normalization to any time point, and results can be directly viewed in the software window. We conducted the normalization at one time point before the treatment.

### 2.14. Extracellular Flux Assay

Mitochondrial function effects of metformin were determined on the XF24 Seahorse Biosciences Extracellular Flux Analyzer (Agilent Seahorse Technologies). Cells growing in regular media were plated at a density of 7500 cells/well onto XFp cell culture miniplates (Seahorse XFp FluxPak, Cat. #103022-100, Agilent Seahorse Technologies) and allowed to adhere overnight at 37 °C with 5% CO_2_ in a humidified incubator. Culture medium was removed from each well and replaced with fresh medium containing 5 mmol/L metformin or vehicle for 24 h. Experimental media were removed and cells were washed and incubated with pre-warmed assay media (XF Base Medium Minimal DMEM containing 10 mmol/L glucose, 1 mmol/L sodium pyruvate, and 2 mmol/L glutamine) for 1 h in a non-CO_2_ incubator at 37 °C. The Seahorse XFp Cell Mito Stress Test Kit (Cat. #103010-100, Agilent Seahorse Technologies) was used to measure OCR and the extracellular acidification rate (ECAR) with sequential treatment with 1.5 μmol/L oligomycin A, 1 μmol/L FCCP, and 0.5 μmol/L rotenone/antimycin. Both OCR and ECAR data were normalized for cell number.

### 2.15. Mitochondrial Function Phenotyping

Mitochondrial activity was measured in triplicate using 96-well MitoPlate S-1 plates (Cat. #14105, Biolog, Hayward, CA). Wells containing the different cytoplasmic and mitochondrial metabolic substrates (*n* = 31) were rehydrated with a solution containing mitochondrial assay solution (MAS) (Biolog cat. #72303), redox dye mix (Biolog cat. # 74353), and 30 μg/mL saponin in sterile water. Cells were washed with PBS and resuspended in 1× MAS and added to each well at a final cell density of 30,000 cells/well. Metabolism of substrates was assessed by monitoring colorimetric change of the terminal electron acceptor tetrazolium redox dye at a wavelength of 590 nm on a kinetic microplate reader. 

### 2.16. Statistical Analysis

For all experiments, at least three independent experiments were performed with *n* ≥ 3 replicate samples per experiment. Data are presented as mean ± S.D. Bar graphs, curves, and statistical analyses were generated using GraphPad PRISM 5.0 (GraphPad Software, Inc., San Diego, CA, USA). Two-group comparisons were performed using Student’s t test for paired and unpaired values. Comparisons of means of ≥3 groups were performed by ANOVA, and the existence of individual differences, in case of significant F values at ANOVA, was tested by Tukey’s multiple contrasts. Statistical tests were two-sided. 

## 3. Results

### 3.1. Metformin Directly Targets SHMT Enzymatic Activity in an Isoenzyme-Selective Manner

We first assessed the differential inhibitory effects of metformin on purified, human recombinant SHMT1 and SHMT2 isoforms using a spectrophotometric assay in which both isoenzymes are incubated with saturating concentrations of glycine, a fixed concentration of formyl-THF and a low concentration range of metformin (Figure 1A). Enzyme activity was calculated by measuring the so-called quinonoid intermediate, which is generated when both glycine and formyl-THF bind the SHMT1/2 enzymes to form an SHMT–glycine–folate ternary complex [18,20,22] and displays an intense absorption band with a maximum at ~500 nm deriving from the deprotonation of glycine (Figure 1A, top). Binding of an inhibitor to the SHMT1/2 enzymes reduces the absorbance at 502 nm [20,21,33]. The estimated apparent inhibition constants of metformin were 100 ± 33 mmol/L for SHMT1 and 39 ± 6 mmol/L for SHMT2 (Figure 1A, bottom). These findings suggest that metformin might directly target the enzymatic activity of SHMTs in an isoenzyme (SHMT2)-selective manner.

### 3.2. Binding of Metformin to SHMT2 Does Not Involve the SHMT2 Ligand-Binding Sites

Given the ability of metformin to preferentially target human SHMT2 over SHMT1, we next characterized the inhibition mechanism of metformin against SHMT2 in more detail. We first measured quinonoid intermediate-related absorbance changes at 502 nm upon addition of varying concentrations of formyl-THF to a solution of SHMT2 using a fixed glycine-saturating concentration and graded concentrations of metformin (Figure 1B, left panels). In parallel experiments, we used the same concentration range of metformin while varying glycine concentrations and maintaining formyl-THF fixed at a saturating concentration (Figure 1B, right panels). All reactions were carried out at pH 7.8 to mirror the alkaline pH of the mitochondrial matrix [34]. For both series, saturation curves and double reciprocal plots were globally fitted to a mixed-type inhibition equation, obtaining significantly different K_i_ values for the binding of metformin to the SHMT2/formyl-THF (13.3 ± 2.1 mmol/L) and SHMT2/glycine (51.2 ± 2.4 mmol/L) binary complexes. Notably higher K_i1_ values (>150 mmol/L metformin) were obtained describing the binding of metformin to the SHMT2/formyl-THF/glycine ternary complex. These findings indicate that metformin binds with higher affinity to the SHMT2 binary complexes than to the ternary complexes. More importantly, since no competitive inhibition by metformin is observed with both glycine and folate, the inhibitory effect of metformin does not involve the SHMT2 ligand-binding sites.

### 3.3. Metformin Is Computationally Predicted to Occupy the Cofactor Pyridoxal 5′-phosphate Cavity of SHMTs

To derive a structure-based rationale for both the observed preferential inhibition of mitochondrial SHMT2 over cytosolic SHMT1 and the predicted binding of metformin at a site other than the ligand-binding sites, we performed in silico structural studies based on docking and molecular dynamics (MD) simulations of metformin with the crystal structures of SHMTs. Specifically, we assessed the ability of metformin to engage cavities that might directly interfere with the oligomeric state of SHMTs and/or to occupy the binding site of the PLP cofactor, which is known to differentially regulate the oligomerization of SHMT1 and SHMT2 [19].

We initially evaluated whether the ability of metformin to interact with the interface surface of SHMT dimers would be sufficiently strong and stable to hinder the formation of a catalytically active tetramer. Although metformin was predicted to occupy (with otherwise low-binding energies) several cavities at the dimer-to-dimer contacts of SHMT enzymes, careful assessment of the trajectory data revealed that all the metformin molecules were predicted to leave their initial docking site before 5 ns (Appendix A). Therefore, even though the interaction energy between dimers of dimers was approximately four times lower than that between the monomers in the dimer, the computational analysis failed to support a model in which metformin might directly disrupt tetramer formation of SHMTs (Appendix A).

We next explored the putative atomic interactions between metformin and the PLP cavity at pH 7.0 and 8.0, to take into account how the catalytic efficiency of SHMT1 and SHMT2 relates to their respective cytoplasmic and mitochondrial matrix microenvironments [34]. Computational scanning predicted a pH-independent capacity of metformin to occupy the PLP pocket at both the ligand-free (apo) and ligand-bound (holo) structures of SHMT1 and SHMT2 (Figure 2). The occupancy was predicted to occur with a relatively good binding energy (−5.0–−6.0 kcal/mol) if one acknowledges the small size of metformin and that docking calculations performed against cavities are generally biased toward the ligand (i.e., PLP) to which the target structure (i.e., SHMT) is co-crystallized. The interaction maps between metformin and SHMT amino acids predicted the occurrence of both hydrogen bonds and hydrophobic interactions involving key catalytic residues including Lys257 for SHMT1 and Lys280 for SHMT2, among others (e.g., Glu98, His171, and Arg425). As Lys257 and Lys280 are known to form a Schiff base in the presence of PLP [35,36], the presence of metformin was predicted to block SHMT enzymatic activity by preventing the entry of PLP to the cavity and, consequently, the formation of the internal aldimine state (Figure 2). Close inspection of the arrangement of metformin on the PLP binding cavity revealed that, in some cases, such a location was deep enough to prevent its visibility on the surface. MD simulations (100 ns) revealed the occurrence of small rearrangements (less than 4 Å) of the metformin position at the PLP binding cavity, which were achieved during the first 5 ns and remained stable over time, with the notable exception of apo-SHMT2, which required approximately 50 ns (Appendix A). The negative solvation energies obtained following molecular mechanics Poisson–Boltzmann (MM/PBSA) rescoring calculations over MD simulations strongly suggested the existence of destabilizing interactions in all the analyzed situations (Appendix A).

### 3.4. Metformin Is Computationally Predicted to Destabilize the Oligomerization of a Catalytically Active SHMT2 Conformation

The binding of PLP promotes the stabilization of the quaternary structure of SHMT2, favoring the formation of tetramers [19]. Conversely, metformin is in silico predicted to favor open dimers over closed ones. To computationally test this hypothesis, we aligned the alpha carbons of the SHMT2/metformin complex after 100 ns of MD simulations with the structurally open SHMT2 dimeric—but not the active PLP-bound tetrameric—structure present in the recently discovered BRISC–apo SHMT2 complex [37,38]. Whereas such an alignment failed to show any significant difference in the case of SHMT1, root-mean-square deviation values notably decreased in the case of apo- and holo-SHMT2 (Appendix A). Metformin is therefore predicted to shift the secondary structure of SHMT2 to one closer to the enzymatically inactive conformation stabilized within the BRISC–SHMT2 complex. We then employed the empirical force field FOLDx [39,40,41] to quantitatively estimate how metformin might impact on the interactions contributing to the stability of SHMT complexes. The presence of metformin markedly decreased the FOLDx-calculated interaction energy values between the monomers of the closed dimers of SHMT2 at the end of the 100 ns MD simulation, with a notably lesser effect in the case of SHMT1 (Appendix A). The in silico approach predicted that the ability of metformin to occupy the PLP cavity might destabilize the formation of catalytically active SHMT2 oligomers. The PLP-driven oligomerization switch does not occur in SHMT1, which constitutively exists as a tetramer in a PLP-independent manner [19], thereby providing a plausible mechanistic scenario to explain the ability of metformin to impair mitochondrial SHMT2 activity while largely sparing the cytosolic SHMT1 one.

### 3.5. Metformin Impairs the Capacity of PLP to Promote SHMT2 Dimer-to-Tetramer Transition

To gain insight into the possible mechanism of action of the inhibitory activity of metformin towards SHMT2, we next employed differential scanning fluorimetry (DSF) to directly evaluate the capacity of metformin to differentially alter the conformational space of SHMT2 (Figure 3). DSF is a rapid and popular screening method in drug discovery to identify low-molecular-weight ligands that bind and stabilize purified proteins, evaluating the “ligandability” of a target protein [42,43,44,45]. Ligands that interact with proteins typically stabilize the folded proteins, leading to a shift in the midpoint of the unfolding transition (i.e., the melting temperature, T_m_). Accordingly, observing changes in the T_m_ as a function of ligand concentration can serve as evidence of a direct interaction. Careful examination of the resulting thermal denaturation profiles revealed that, in response to PLP, SHMT2 (but not SHMT1) showed a very different stability curve in the presence of metformin (Figure 3A). In the case of SHMT1, the K_d_ for PLP binding was unaltered by the presence of metformin (Figure 3B, *top*
*left*). By contrast, we observed a metformin-induced 6-fold decrease in the binding affinity of PLP to apo-SHMT2, with the K_d_ towards apo-SHMT2 increasing from 0.3 μmol/L in the absence of metformin to 1.8 μmol/L in its presence (Figure 3B, *top*
*right*).

DFS-based approaches examining the mode of binding of metformin to SHMT1 and SHMT2 supported the computational prediction that metformin impairs the capacity of PLP to promote a dimer-to-tetramer transition of SHMT2. The observed loss of thermal stability of SHMT2 but not of SHMT1 strongly suggests that the presence of metformin hinders the capacity of PLP to promote the conversion from an inactive open state to a highly ordered, catalytically competent closed state, a switch of the oligomeric state that occurs only in SHMT2 and not in SHMT1 (Figure 3B, *bottom*).

### 3.6. Metformin Blocks SHMT2 Activity in Cancer Cells

To validate the capacity of metformin to specifically engage SHMT2 and block its activity in cells, we first measured SHMT activity in living cells using a radioisotope assay based on the ability of SHMT to catalyze the exchange of the pro-2S proton of glycine with solvent [21,46] (Figure 4A). We utilized HAP1 cells, a human haploid cell line that has been used previously as a model for SGOC metabolism [10,17,33]. In parallel, we employed paired SHMT1 knockout (KO)- and SHMT2 KO-HAP1 cells produced by CRISPR/CAS9-mediated frameshift deletions of *shmt1* and *shmt2* mRNAs in exons 5 and 2, respectively. Metformin treatment of HAP1 parental cells caused a partial but significant decrease in SHMT activity at low metformin concentrations and full suppression of SHMT activity at higher concentrations (Figure 4A, *left*). The 50% inhibitory concentration of metformin for SHMT activity in HAP1 parental cells (IC_50_ = 31 mmol/L) was marginally but not significantly modified in SHMT1 KO-HAP1 cells (IC_50_ = 38 mmol/L; Figure 4A, *right*). By contrast, significantly higher concentrations of metformin were needed to achieve the IC_50_ in SHMT2 KO-HAP1 cells (~60 mmol/L), and SHMT activity was not fully suppressed in SHMT2 KO-HAP1 cells even at metformin concentrations of 150 mmol/L (Figure 4A, *right*). Loss of SHMT2 promoted resistance to the anti-SHMT activity of metformin not only in terms of the IC_50_ but also by reducing its maximum inhibitory effect in living cells, confirming and extending the spectrophotometric enzymatic assays with purified SHMTs. 

The radioisotopic approach demonstrated that metformin is an isoenzyme-selective inhibitor of SHMTs that directly blocks mitochondrial SHMT2 in cancer cells. However, we acknowledge that the selective inhibitory activity of metformin towards SHMT2 activity based on spectrophotometric, DSF, and radioisotopic assays took place in a mid-millimolar range of non-physiological metformin doses.

### 3.7. Metformin Reduces the SHMT2-Channeled Serine-to-Formate Flux

Using SHMT1 KO-HAP1 cells to render them uniquely dependent upon mitochondrial SHMT2 enzymatic activity and prevent metabolic compensation by reversal of SHMT1 catalysis (serine → glycine) in response to loss of SHMT2 [47], we decided to preliminarily explore if more physiologically relevant concentrations of metformin could somewhat impact the mitochondrial serine catabolism in cancer cells by culturing cells in the presence of the [U-^13^C]serine tracer and determining its catabolization to extracellular formate (Figure 5A, [32]). In this context, M + 0 formate reflects the formate derived from de novo synthesized serine from glucose, whereas M + 1 formate reflects the formate derived from the imported, extracellular [U-^13^C]serine (Figure 5A). Treatment with typically used concentrations of metformin required to induce anti-cancer effects in in vitro studies (5–10 mmol/L)—a range of concentrations that might be achieved in the mitochondrial matrix upon heavy accumulation of metformin due to the mitochondrial membrane potential [16,48,49]—notably reduced the release of M + 0 and M + 1 formate in SHMT1 KO-HAP1 cells (Figure 5A). Given the ability of metformin to disrupt the mCI-dependent functioning of the NAD-dependent methylenetetrahydrofolate dehydrogenase/cyclohydrolase MTHFD2 [32], this finding might merely reflect a reduction of formate release due to an impaired mCI activity.

The β-carbon (3-carbon) of serine serves as the major 1C donor in proliferating cancer cells. Serine enters the mitochondria, and its hydroxymethyl group is released from the mitochondria as formate. Serine labeled with ^13^C at carbon 3 transfer ^13^C to 5,10-CH_2_-THF only via the SHMT activity, whereas glycine cleavage would donate carbon 2 (^12^C) of serine, thereby excluding any putative carbon contribution from a functional glycine cleavage system. We therefore decided to trace how metformin treatment impacted endogenous formate utilization by monitoring the metabolic fate of serine 3-carbon using labeling experiments with [3-^13^C]serine as the tracer ([32,50,51,52]; Figure 5B). We additionally tested whether metformin could regulate the mitochondrial SHMT2-channeled serine-to-formate flux beyond its reported capacity to uncouple mCI activity from MTHFD2 functioning [32] by overexpressing the metformin-resistant *Saccharomyces cerevisiae* NADH dehydrogenase NDI1, which can correct functional defects in mCI [16].

Metformin treatment markedly increased the release of M + 0 and M + 1 formate in SHMT1^+^/SHMT2^+^ HAP1 parental cells (Figure 5B). This finding might reflect a reversal of the (SHMT1-dependent) cytosolic 1C flux from serine-production to serine-catabolism to compensate for the block in mitochondrial formate production by metformin [32,47]. Indeed, the metformin-augmented release of formate was largely, but not entirely, reversed in HAP1 cells engineered to overexpress the metformin-resistant NADH dehydrogenase NDI1 (Figure 5B). We hypothesized that if the channeling of serine-derived carbon to formate occurs exclusively via mitochondrial SHMT2, then metformin treatment should result in a small but significant inhibition of formate release independently of mCI. To prevent interference from compensatory cytoplasmic folate reactions, we repeated the experiments in SHMT1 KO-HAP1 cells, finding that metformin treatment likewise inhibited the amounts of released M + 0 and M + 1 formate to a similar extent (Figure 5B). These results indicate that the ability of metformin to target serine-to-formate metabolism does not involve changes in the synthesis of serine from glucose, but rather an impairment of mitochondrial serine catabolization. We then re-assessed how metformin treatment impacts formate release in SHMT1 KO-HAP1 cells engineered to overexpress the metformin-refractory yeast NDI1 protein. Although NDI1 overexpression largely circumvented the capacity of metformin to block mitochondrial serine catabolization, it was noteworthy that metformin retained its ability to reduce, by up to 30%, the mitochondrial-dependent formate release (Figure 5B). Moreover, metformin treatment significantly reduced the incorporation of serine-derived 1C to deoxythymidine triphosphate (dTTP) in SHMT1-KO HAP1 cells even upon overexpression of the metformin-refractory yeast CI (NDI1) (Figure 5C). Indeed, radioisotope assays confirmed that NDI1 overexpression cells failed to alter the ability of metformin to directly engage and block the mitochondrial activity of SHMT2 in living cancer cells (*data not shown*). Metformin treatment drastically augmented the concentration of the purine precursor 5-aminoimidazole-4-carboxamide ribonucleotide (AICAR) in SHMT1 KO-HAP1 cells (Figure 5C), thereby phenocopying the elevation of AICAR occurring in cells deficient of SHMT2 or treated with the SHMT inhibitor SHIN1 [4,47,55,56]. The ability of metformin to significantly increase the contribution of 1C to S-adenosylmethionine (SAM)—the universal methyl donor for cellular methylation [10]—was fully blunted in SHMT1-KO HAP1 cells irrespective of mCI (Figure 5C). 

These findings, altogether, suggest that metformin, upstream of the metformin-targeted mCI/MTHFD2 functional coupling, could partially curtail serine-to-formate catabolism by directly inhibiting SHMT2 in cancer cells.

### 3.8. Loss of SHMT2 Prevents the Cytotoxic and Anti-Proliferative Effects of Metformin

The metabolic synthetic lethality between metformin and serine starvation phenocopies the strong cell growth defects occurring in SHMT2 (but not in SHMT1)-KO cells growing under serine withdrawal [57]. We therefore envisioned that the anti-proliferative effects of metformin would be affected by the loss of its putative SHMT2 target in cancer cells. Cell viability assays revealed that SHMT2-KO HAP1 cells were significantly refractory to the cytotoxic effects of metformin (IC_50_ = 47.4 mmol/L) when compared with their HAP1 parental counterparts (IC_50_ = 14.2 mmol/L) (Figure 6A). This acquired resistance to metformin was completely rescued upon restoring SHMT2 protein expression in SHMT2-KO HAP1 cells engineered to overexpress an SHMT2 cDNA (Figure 6A). The use of the xCELLigence^®^ RTCA DP instrument, which employs noninvasive electrical impedance monitoring to quantify cell proliferation in a label-free, real-time manner, confirmed that SHMT2 KO-HAP1 cells were largely unresponsive to metformin, whereas SHMT2 reintroduction rescued the anti-proliferative activity of metformin (Figure 6B).

### 3.9. Mitochondrial Serine Catabolism Is a Determinant for the Anti-Cancer Activity of Metformin

As the cancer inhibitory effects of metformin are cancer cell-autonomous and depend on its ability to inhibit mCI, the refractoriness of SHMT2-KO HAP1 cells to metformin might reflect the requirement for SHMT2 for proper mitochondrial metabolic functioning including the assembly of mCI [57,58,59]. Accordingly, we observed a drastic reduction of the basal oxygen consumption rate (OCR) and of the capacity of SHMT2 KO-HAP1 cells (but not of SHMT1 KO-HAP1 cells) to maximize their rate of respiration in response to the uncoupling agent FCCP (Figure 7). SHMT2 KO-HAP1 cells remained sensitive to the inhibitory effects of metformin on maximal OCR, ATP production, and coupling efficiency (Figure 7).

When we examined the capacity of SHMT2-containing and SHMT2-less mitochondria to metabolize 31 cytoplasmic and mitochondrial metabolic substrates in a 96-well assay (MitoPlate S-1 plates) in saponin-permeabilized HAP1 and SHMT2 KO-HAP1 cells, we observed that loss of SHMT2 fully impeded the generation of reducing equivalents from numerous tricarboxylic acid (TCA) cycle substrates (Figure 8). Conversely, a noteworthy augmentation of TCA cycle substrates utilization was observed in SHMT1 KO-HAP1 cells (Figure 8). 

Overall, these findings suggest that, while partially maintaining its capacity to inhibit mitochondrial OXPHOS activity, metformin loses its cytotoxic and antiproliferative activity in SHMT2-null cancer cells unable to produce energy-rich NADH or FADH_2_ molecules from TCA cycle substrates. 

## 4. Discussion

Despite the intense effort by many investigators in the last decade, the detailed molecular targets mediating the anti-cancer effects of the anti-diabetic biguanide metformin remain unclear [60]. We here define metformin as a direct and specific small-molecule inhibitor of mitochondrial SHMT2. Mechanistically, metformin appears to interfere with PLP binding and, therefore, with the conformational change of SHMT2 toward a closed dimer and the consequent active tetramer [19]. This ability of metformin to curtail the capacity of the cofactor PLP to promote the conversion of SHMTs from an inactive open state to a highly ordered, catalytically competent closed state, involves an oligomerization switch that solely occurs in mitochondrial SHMT2 and is absent in cytosolic SHMT1 [19]. As SHMT1 constitutively adopts a PLP-independent tetramer conformation, metformin treatment specifically inhibits the activity of mitochondrial SHMT2 while sparing the closely related activity of cytosolic SHMT1. From a medicinal chemistry perspective, our data offer new structural and mechanistic insights as to how biguanides such as metformin could be improved to produce more potent and specific PLP-competitive inhibitors of the SHMT2 activating oligomerization. From a physiological and therapeutic perspective, our data might suggest a new mechanism of metformin action that, although requiring very high concentrations of the drug, would become operative when metformin facilitates the so-called “methyl folate trap” that is known to restrict the functioning of SHMT enzymes. 

Among the central enzymes involved in SGOC metabolism, SHMT remains as one of the few for which an established therapeutic agent has not yet been developed. Cytosolic SGOC enzymes such as dihydrofolate reductase (DHFR) and thymidylate synthase (TS) have been important targets for pioneering anti-folates such as aminopterin, methotrexate, pemetrexed and 5-fluorouracil [61,62]. Although there has been an ever-increasing focus on exploiting the essential role of mitochondrial SGOC enzymes as critical sources of 1C units to sustain the malignant phenotype in tumors, small-molecule inhibitors of the first (SHMT2) and second (5,10-methylene tetrahydrofolate dehydrogenase 2 (MTHFD2)) mitochondrial SGOC enzymes have not yet reached the clinic [2,63,64,65]. Whereas the search for selective serine analogues and amino acid derivatives as SHMT2 inhibitors has not been very successful, the pyrazolopyrans scaffold represents the leading series of compounds accounting for the SHMT1/2 inhibitor space. Accordingly, folate-competitive/folate mimetic cell-permeable inhibitors of SHMT2 such as SHIN2 and AGF347 have showed in vivo target engagement at nanomolar–midmicromolar concentrations notably lower than those required for FDA-approved antifolates to inhibit SHMTs. Using a repurposing approach, the antidepressant sertraline has been shown to suppress serine/glycine synthesis by potentially binding the exact same pocket as the dual SHMT1/2 inhibitor SHIN1 [66]. Using a novel fluorescent assay measuring SHMT2 activity directly, the so-called “hit 2” has recently been reported as a non-(serine) competitive, selective inhibitor of SHMT2 [67]. Intriguingly, despite the well-accepted notion that SHMT2 could be inhibited by displacing PLP with a stronger-binding molecule [38], our capacity to hinder how PLP binding stabilizes the active tetrameric state of SHMT2 has not been particularly successful to date. Our current findings revealing that metformin can occupy the PLP cavity and destabilize the formation of catalytically active SHMT2 oligomers might be valuable for the development and discovery of better PLP cavity-targeted agents capable of specifically decreasing SHMT2 activity.

As SHMT2 is essential not only in cancer pathology, but also in normal physiology, the mitochondria-specific inhibition of SHMT2 activity remains challenging [68,69]. However, one should acknowledge that anti-cancer models that disrupt SHMT2 as a proxy for drug treatment likely overestimate the undesirable impact of SHMT2 targeting. Thus, whereas SHMT2 deletion will fully inhibit mitochondrial translation, SHMT2-targeted drugs might produce relevant anti-cancer effects without achieving the full depletion of mitochondrial 1C units [59]. Supporting our current proposal that metformin might operate as a small-molecule inhibitor of SHMT2, metformin has recently been reported to mimic the ability of serine catabolism blockade to ameliorate disease progression in a murine model of neuromuscular Leigh syndrome [70]. Thus, rather than aggravating the existing mCI deficiency in mice lacking the mCI subunit NDUFS3, metformin phenocopied the pharmacological inhibitor of SHMT1/2 SHIN2 to attenuate the so-called clasping phenotype [70,71]. Further studies are warranted to better understand the molecular mechanisms underlying the phenotypic outcomes arising from single and combined treatments with weak (millimolar) regulators of SHMT2 activity impeding a dimer-to-tetramer transition of SHMT2 in a PLP-dependent manner (e.g., the biguanide metformin) and potent (nanomolar) SHMT2 inhibitors directly occupying the folate binding site of SHMT2 such as SHIN-like optimized pyrazolopyrans or sertraline [4,63,64,65,66]. Intriguingly, the SHMT1/2 inhibitor SHIN1 has been found to notably augment cancer cell proliferation in the presence of the mCI inhibitors metformin, phenformin, and rotenone [71]. Thus, when respiration is impaired, excessive NADH generation via serine catabolism has been proposed to decrease cell growth [71]. Conversely, the antiproliferative activity of the SHMT1/2 inhibitor sertraline was further potentiated in the presence of suboptimal dosages of mitochondrial inhibitors rotenone and antimycin A, targeting mCI and III, respectively [66]. The latter findings have suggested that simultaneous inhibition of SHMT1/2-driven serine metabolism and mitochondrial functioning can be a novel treatment strategy for serine/glycine synthesis-addicted cancers [66]. While we remain cautious about the therapeutic relevance of the anti-SHMT2 mechanism of action in the context of supra-physiological concentrations (>1 mmol/L) of metformin, we acknowledge that additional sets of experimental approaches (e.g., usage of [2,3,3-^2^H_3_] to trace the relative contribution of the cytosolic (SHMT1) and mitochondrial (SHMT2) 1C metabolism) should be considered to unambiguously determine the partitioning ability of a putative, dual mCI-SHMT2 inhibitor such as metformin between cytosolic and mitochondrial fluxes of folate-mediate serine oxidation.

While numerous parallelisms have been noted between the cellular responses to metformin and those to anti-folate drugs such as methotrexate, most have been ascribed to the action of metformin on host–gut microbiota interactions involving inhibition of bacterial folate and methionine metabolism (e.g., *Escherichia coli* DHFR; [72,73]), and less consideration has been given to the direct inhibitory activity that metformin might exert on SGOC metabolic enzymes in human cancer cells. Our demonstration of metformin as a PLP-competitive inhibitor of SHMT2 oligomerization might open new horizons for understanding the anti-cancer behavior of metformin. In particular, it might provide an additional dimension to the well-reported phenomenon of vitamin B12 depletion occurring upon metformin treatment [74,75,76,77]. Metformin-driven B12 deficiency might result in the “trapping” of a pool of methyl-THF, thereby causing an intracellular shortage of biologically active THF [78,79] (Figure 9, *top panel*). In such a scenario of secondary folate deficiency, the accumulation of methyl- and formyl-THF, which also bind and inhibit SHMT enzymes [80], could exacerbate the inability of SHMT2 to operate in the presence of the PLP-competitive inhibitor metformin (Figure 9, *top panel*). The SGOC-related anti-cancer actions of metformin, which should not exclusively be viewed as only a downstream consequence of mCI targeting, but also as a causal upstream inhibitory effect directly involving SHMT2-driven mitochondrial serine catabolism, might illuminate an unforeseen capacity to synergize or antagonize with other interventions involving SHMT1 and/or SHMT2 functioning [10,53,58,59,81,82,83]. The ability of metformin to either improve or impede the therapeutic response of tumor cells to serine and glycine starvation [11,12] might also be reconsidered, at least in part, in terms of the direct SHMT2-targeted effects of metformin.

The inactive apo-SHMT2 dimer has recently been recognized as an unexpected driver of immune signaling capable of promoting inflammatory cytokine production via interaction with the deubiquitylating BRCC36 isopeptidase complex (BRISC) [37,84]. As the active PLP-bound tetramer fails to bind and inhibit BRISC, the intracellular levels of PLP ultimately determine the inflammatory cytokine responses by eliciting or impeding the interaction between SHMT2 and BRISCR. Accordingly, natural and synthetic SHMT2 binders such as folate and PLP analogs can alter the availability or conformation of dimeric SHMT2 and, consequently, inflammatory signaling [36,37]. Future studies are needed to evaluate whether PLP availability might cooperate with the ability of metformin or metformin-based derivatives to promote a more disordered state of apo-SHMT2 that is less prone to PLP-induced SHMT2 oligomerization, might generate a non-enzymatic, metformin-bound SHMT2 dimer unable to interact with the BRISC complex and, therefore, with anti-inflammatory and immune-regulatory effects (Figure 9, *bottom*
*panel*).

## 5. Conclusions

Folate-dependent SGOC metabolism contributes to core cellular building blocks (e.g., purine and pyrimidine synthesis), epigenetics, post-translational modifications, and redox homeostasis in cancer cells, thereby representing a key metabolic hallmark of the oncogenic phenotype [85,86,87]. As currently available SHMT2 inhibitors have not yet reached the clinic, our data offer unforeseen mechanistic insights as to how biguanides such as metformin could be improved to produce more potent and specific PLP-competitive inhibitors of the SHMT2 activating oligomerization. 

## Figures and Tables

**Figure 1 cancers-13-04009-f001:**
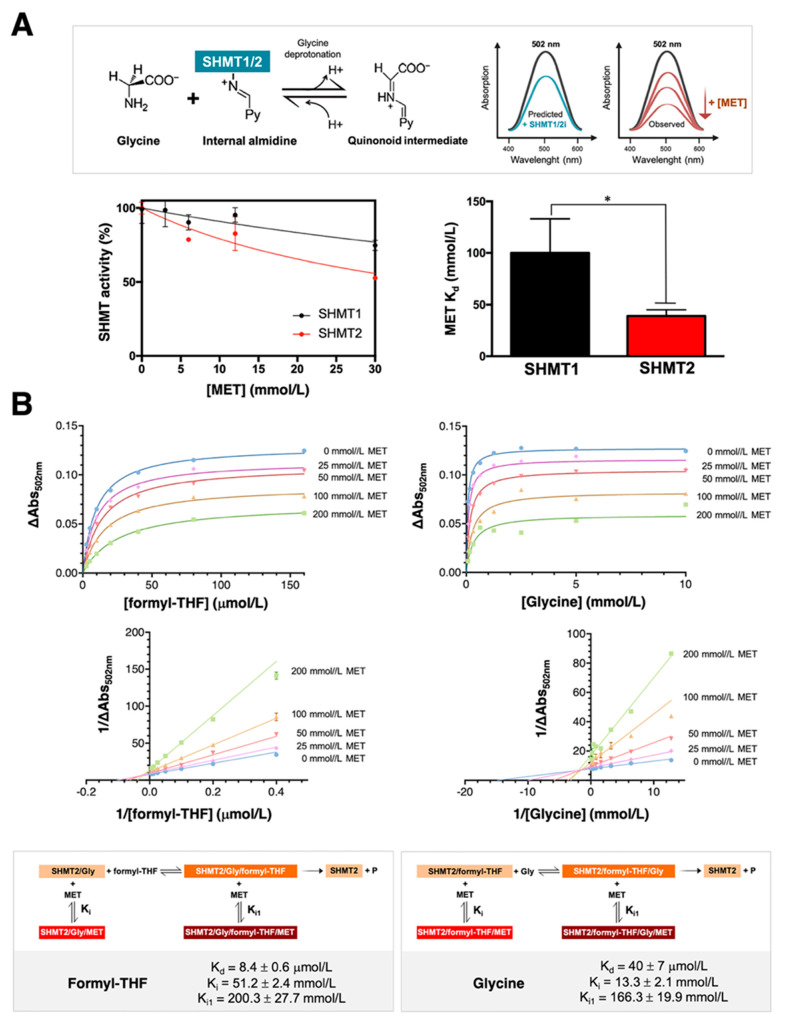
**Metformin inhibits the catalytic activity of human purified SHMT2.** (**A**) *Top*. The quinonoid intermediate formed upon binding of glycine to SHMTs accumulates only when a folate ligand is simultaneously bound to the enzyme, thereby forming an enzyme–glycine–folate ternary complex. In the scheme, the pyridine ring of the cofactor is indicated by Py. *Bottom*. The activity of purified (5 μmol/L) recombinant SHMT1 (black symbols) and SHMT2 (red symbols) was measured in the presence of 5 μmol/L formyl-THF, a saturating concentration of glycine (3 mmol/L) and increasing concentrations of metformin (MET) (0–30 mmol/L). Values are the mean ± standard deviation (SD) of three independent experiments. * *p* < 0.05 (**B**) *Top panels*. Absorbance changes at 502 nm were measured upon the addition of graded concentrations of metformin (0–200 mmol/L) to a solution of 5 μmol/L SHMT, glycine, and formyl-THF. In parallel experiments (*left panels*), assays were performed varying the concentration of formyl-THF (2.5–160 μmol/L) at a fixed glycine concentration (10 mmol/L) or varying the concentration of glycine (0.04–10 mmol/L) at fixed formyl-THF concentration (160 μmol/L). *Bottom panels*. Double reciprocal plots of absorbance changes at 502 nm (plotted as 1/ΔAbs_502 nm_). Kinetic mechanisms of inhibition by metformin are shown in which the SHMT2 enzyme (either the SHMT2–glycine or the SHMT2–formyl–THF complex) forms a complex with metformin according to a variety of dissociation rate constants (K_d_ = K for glycine or formyl-THF binding; K_i_ = K_d_ for metformin binding to the SHMT2 binary complex; K_i1_ = Kd for metformin binding to the SHMT2 ternary complex).

**Figure 2 cancers-13-04009-f002:**
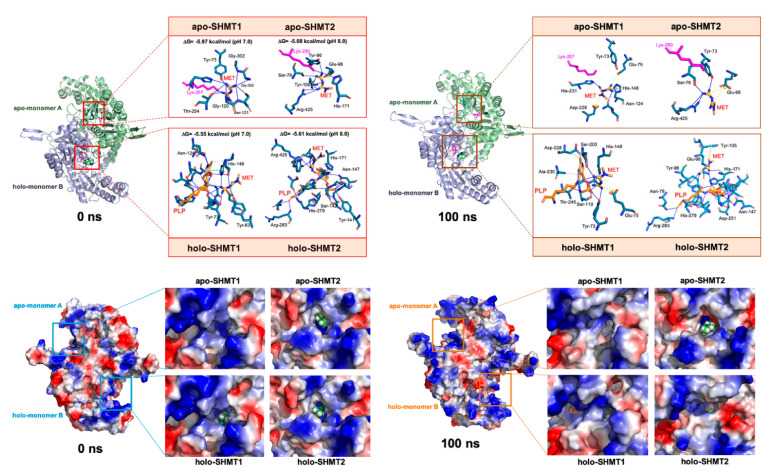
**Metformin is computationally predicted to target the PLP-binding cavity of SHMT.***Top panels.* Models of incorporation of the predicted PLP binding cavity-targeting metformin behavior in human SHMT1 and SHMT2. Figures show the composition of a closed apo-SHMT (monomer A, pale green) and holo-SHMT (monomer B, light blue) before (0 ns) and after (100 ns) a molecular dynamics simulation in which a docked metformin (MET) molecule (represented as spheres) is included in the PLP binding cavity of both subunits. Insets show the detailed interactions of the best poses of MET docked to the PLP-binding site of SHMTs using the PLIP algorithm, indicating the participating amino acids involved in the interaction and the type of interaction (hydrogen bonds, hydrophilic interactions, salt bridges, Π-stacking, etc.). Insets in the left panel include also the Gibbs free energy variation (ΔG, kcal/mol) values calculated from molecular docking experiments (pH 7.4) using YASARA v19.9.17 software. *Bottom panels.* Surface representation of closed SHMT dimers before (0 ns) and after (100 ns) a molecular dynamics simulation. Proteins are represented as a function of the hydrophobicity of its surface amino acids and the Na^+^ and Cl^−^ ions have been eliminated to facilitate visualization. Insets show a detailed view of the PLP binding cavity in SHMT1/2 enzymes. A MET molecule (sphere) is shown when visible from outside the cavity. Figures were prepared using PyMol 2.3 software.

**Figure 3 cancers-13-04009-f003:**
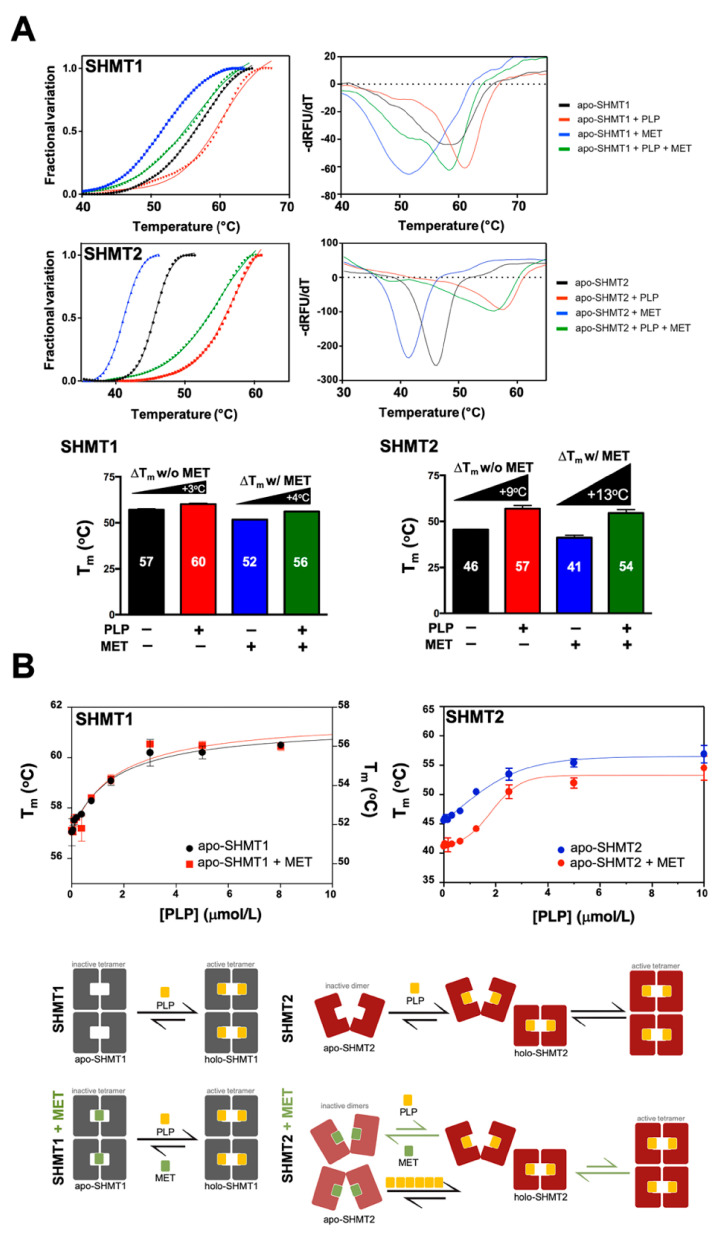
**Metformin regulates SHMT2 dimer–tetramer transition.** (**A**) *Top.* Differential scanning fluorimetry (DSF) analysis of apo-SHMT1 and apo-SHMT2 (0.5 μmol/L each) in the absence and presence of PLP (10 μmol/L) and/or MET (200 mmol/L). *Bottom.* Melting temperatures (T_m_) for each of the conditions shown in left panels. Changes in melting temperatures (ΔT_m_) were calculated by subtracting the T_m_ of SHMT1/2 with buffer and without PLP from the T_m_ of SHMT1/2 with added PLP in the absence and presence of MET. (**B**) *Top*. T_m_ of apo-SHMT1/2 with increasing concentrations of PLP (0–10 μmol/L) in the absence and presence of MET. *Bottom.* Computational- and DSF-based conceptual model of the ability of metformin (MET) to differentially impact the SHMT2 dimer–tetramer equilibrium in response to PLP while sparing the SHMT1 PLP-independent constitutive tetramer.

**Figure 4 cancers-13-04009-f004:**
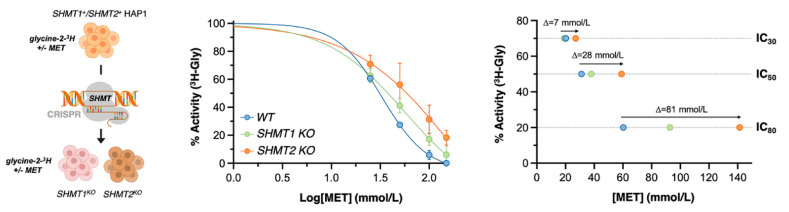
**Metformin selectively inhibits SHMT2 activity in cancer cells.** Suspensions of SHMT1^+^/SHMT2^+^ HAP1 parental cells and SHMT1/SHMT2 KO-HAP1 derivatives were incubated with graded concentrations of metformin (MET, 0–150 mmol/L) for 1.5 h before adding the glycine radioisotope glycine-2-^3^H. *Left.* The results describe the dose-dependent effect of MET on the ability of SHMT to catalyze the exchange of glycine 2-pro-S proton with the solvent. *Right.* The inhibitory effect of MET on SHMT activity occurred in a dose-dependent manner but the IC_30_, IC_50_, and IC_80_ curves were shifted to the right (less potent) in SHMT2 KO-HAP1 cells.

**Figure 5 cancers-13-04009-f005:**
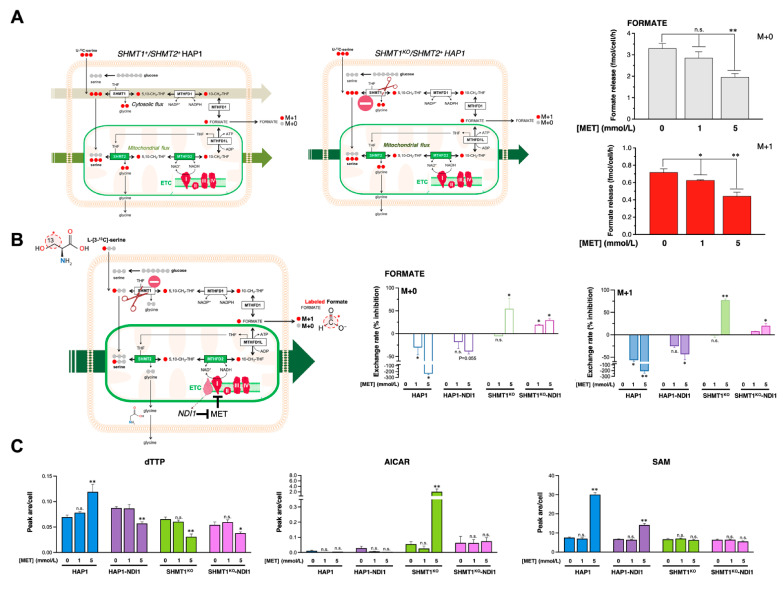
**Metformin targets the mitochondrial serine catabolism via mitochondrial complex I and SHMT2.** (**A**,**B**) Models summarizing mammalian SGOC metabolism in which the oxidation of the third carbon of serine to formate relies on functional mitochondria in a complex I-dependent manner [17,32,52]. Formate produced via mitochondrial SHMT2 is released into the cytosol where it supplies the 1C demand for nucleotide synthesis [53]. Formate can be also recycled back to re-synthesize serine via cytosolic 1C metabolism [47]. Thus, in cells with defective mitochondrial 1C metabolism, the cytosolic pathway is reverted, compensating for the loss of mitochondrial formate production. The metformin-resistant *Saccharomyces cerevisiae* protein NDI1 is a single-subunit NADH dehydrogenase that oxidizes NADH similar to the multi-subunit mammalian complex I without proton pumping or reactive oxygen species generation [16,54]. Graphs show formate release (**A**) and exchange rates of formate (**B**) M + 0 and M + 1 isotopologues in the extracellular milieu of SHMT1^+^/SHMT2^+^ HAP1, SHMT1 KO-HAP1 cells, and isogenic derivatives engineered to overexpress NDI1, all cultured for 24 h in the absence or presence of graded concentrations of metformin (MET). (**C**) Changes in dTTP, AICAR, and SAM following treatment of SHMT1^+^/SHMT2^+^ HAP1, SHMT1 KO-HAP1 cells, and isogenic derivatives engineered to overexpress NDI1 with 1 and 5 mmol/L MET. Data are presented as means (*columns*) ± SD (*bars*) (*n* = 3 cultures representative of at least two independent experiments). Comparisons of means were performed by ANOVA. *p* values < 0.05 and < 0.005 (versus untreated controls) were considered to be statistically significant (denoted as * and **, respectively; n.s. not significant).

**Figure 6 cancers-13-04009-f006:**
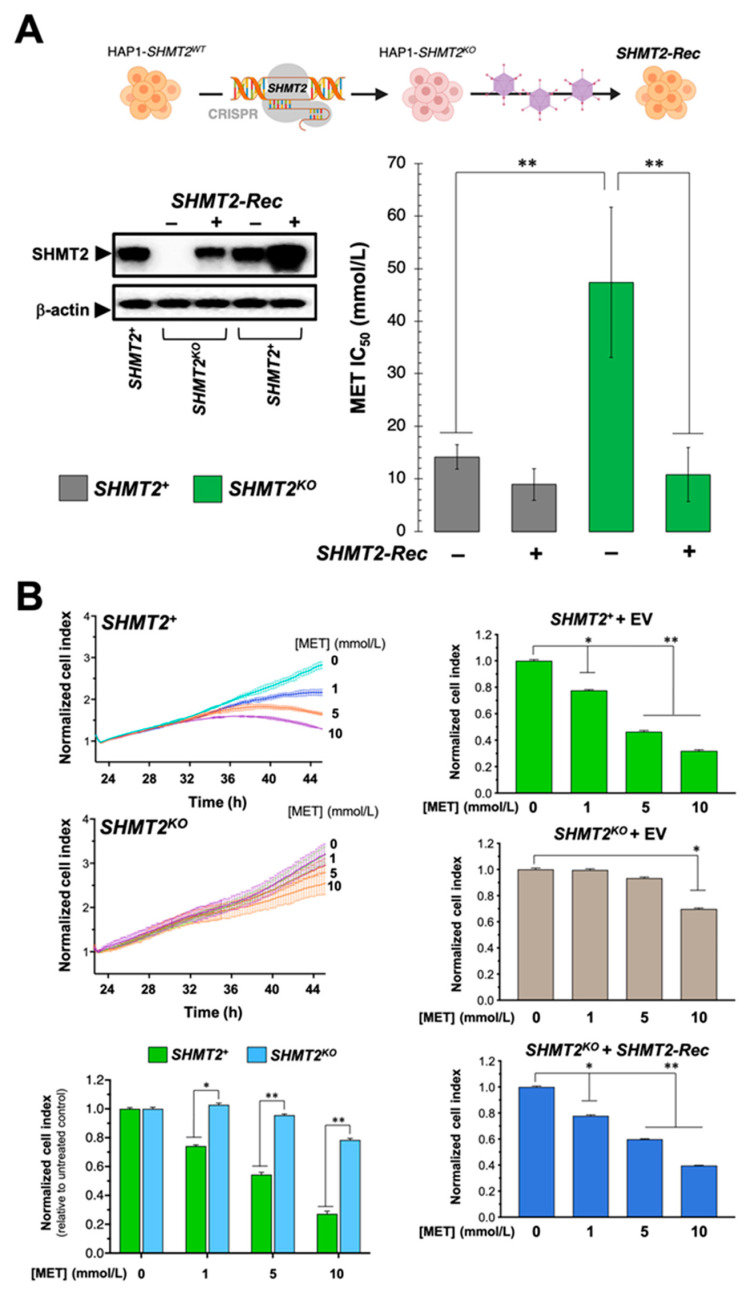
**SHMT2 depletion prevents the cytotoxic and anti-proliferative effects of metformin.** (**A**) *Left.* Representative immunoblot for SHMT2 or β-actin from cell lysates of HAP1 and SHMT2-KO HAP1 cell lines engineered to overexpress SHMT2 cDNA. *Right.* Bar graph showing MTT-based IC_50_ values for metformin in HAP1 and SHMT2-KO HAP1 cells before and after re-expression of SHMT2. Columns are means and error bars are SD. *n* = 3 biological replicates in technical triplicates. (**B**) RTCA profiles and normalized cell indexes of HAP1 and SHMT2-KO HAP1 cells before and after re-expression of SHMT2 cultured in the absence or presence of graded concentrations of metformin (MET). Columns are means and error bars are SD. *n* = 3 biological replicates in technical triplicates. Comparisons of means were performed by ANOVA. *p* values < 0.05 and < 0.005 were considered to be statistically significant (denoted as * and **, respectively; n.s. not significant). EV: empty-vector; Rec: reconstituted.

**Figure 7 cancers-13-04009-f007:**
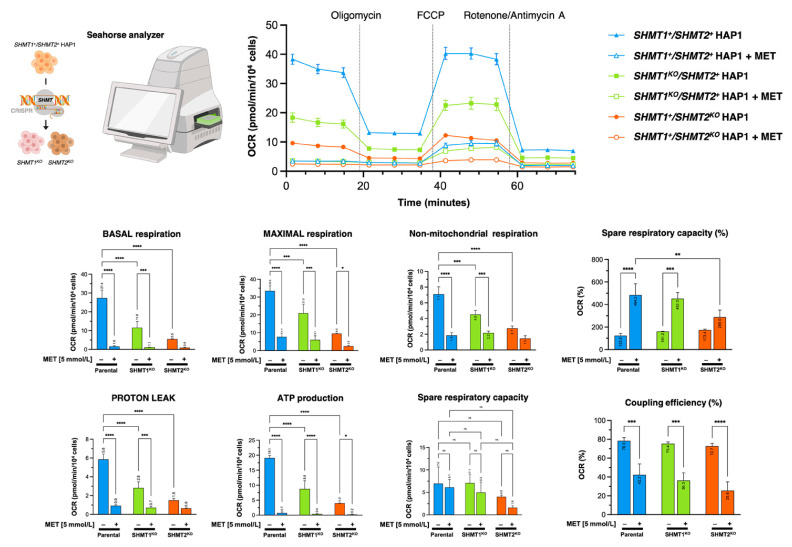
**Depletion of SHMTs alters the mitochondrial bioenergetic profile in HAP1 cells.***Top.* Mitochondrial phenotyping of HAP1 cells upon SHMT1 and SHMT2 loss using the Seahorse XFp Cell Mito Stress Test Assay. *Bottom. OCR profiling.* Seahorse OCR bioenergetic profiles before and after injections of oligomycin, FCCP, and antimycin A in HAP1 parental, SHMT1-KO HAP1, and SHMT2-KO HAP1 counterparts pre-incubated in the absence or presence of 5 mmol/L metformin (MET) for 24 h. **** *p* <0.0001; *** *p* < 0.0005; ** *p* < 0.005; * *p* < 0.05.

**Figure 8 cancers-13-04009-f008:**
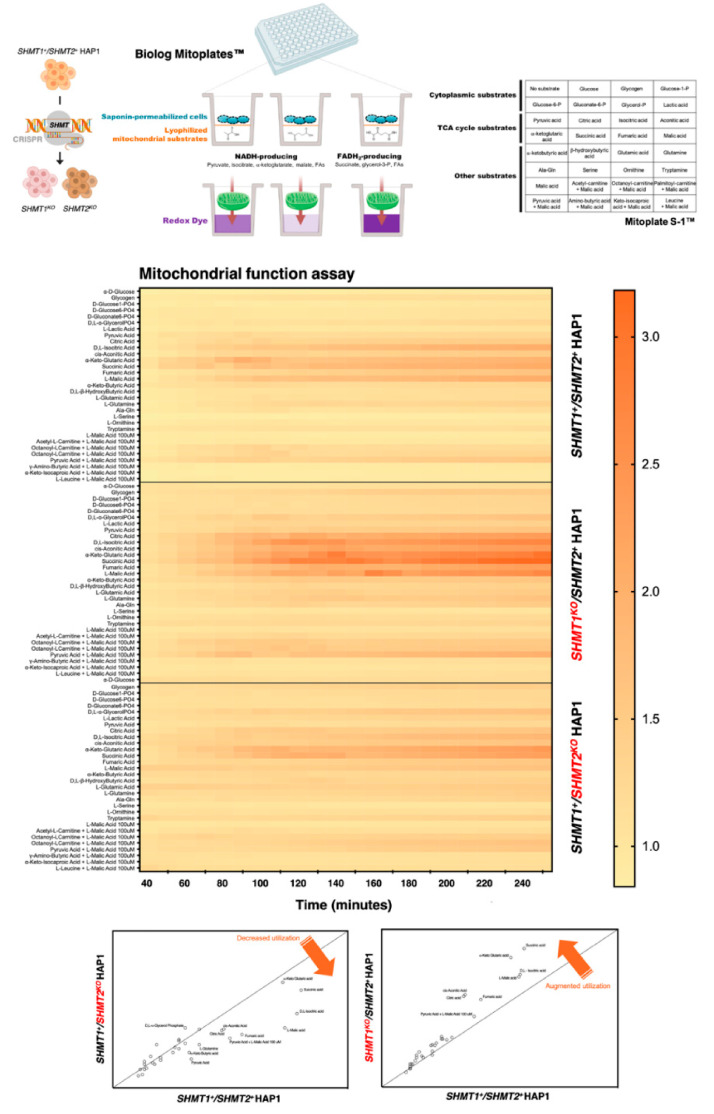
**Depletion of SHMTs alters the mitochondrial functioning in HAP1 cells***Top.* Mitochondrial phenotyping of HAP1 cells upon SHMT1 and SHMT2 loss using Mitochondrial Function Assays with Biolog Mitoplates. Mitochondrial function in HAP1 parental and SHMT2-KO HAP1 counterparts was assayed by measuring the rates of electron flow into and through the electron transport chain (ETC) for cytoplasmic, TCA cycle, and other metabolic substrates producing NAD(P)H (pyruvate, isocitrate, α-ketoglutarate, malate, fatty acids) or FADH_2_ (succinate, glycerol-3-phosphate, fatty acids). The electrons travel from the beginning (complex I or II) to the distal portion of the ETC where a tetrazolium redox dye (MC) acts as a terminal electron acceptor that turns purple upon reduction. *Bottom.* Figure shows the heatmap of the metabolic substrate consumption of fatty acids, glycolysis, amino acids, and TCA cycle metabolites in HAP1 and SHMT2-KO HAP1 cells. Changes in pathway-specific consumption reveal that SHMT2-KO HAP1 cells consumed notably less TCA substrates than HAP1 parental cells.

**Figure 9 cancers-13-04009-f009:**
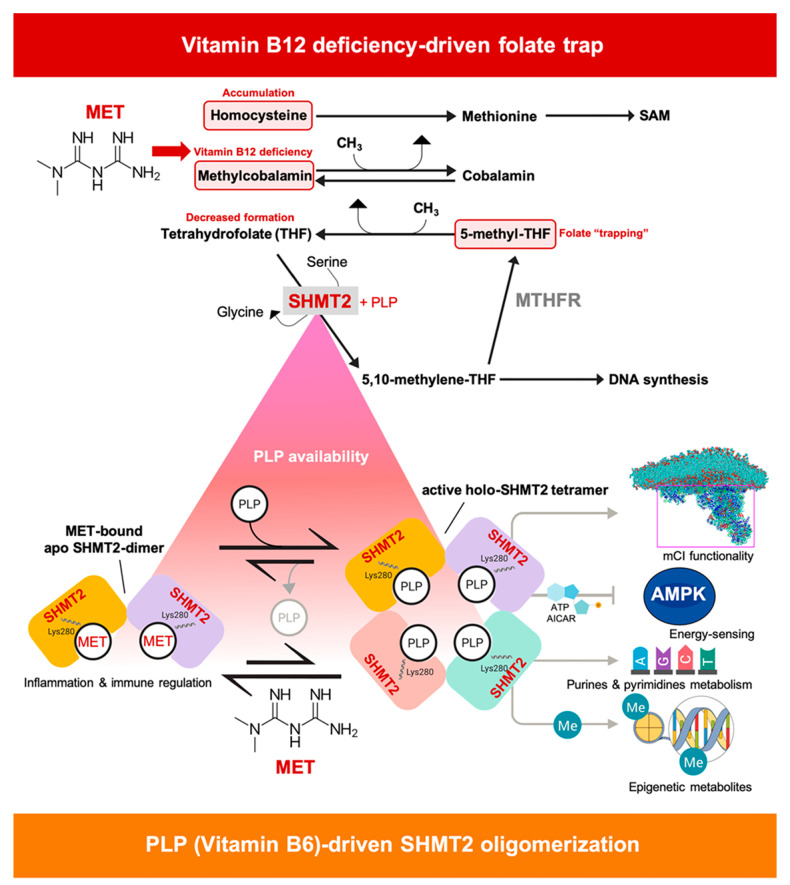
**Metformin and SHMT2 functioning: a working model.** The pharmacological capacity of metformin to operate as a PLP-competitive inhibitor of the oligomeric state of mitochondrial SHMT2 might offer a new perspective for understanding the anti-cancer mechanism of action of metformin. First, mitochondrial SHMT2 is the key channeler of serine-derived carbon into the cytoplasmic and mitochondrial 1C pools of THF-based cofactors that are required not only for nucleotide synthesis but also for mitochondrial respiration, including assembly of the mCI, tRNA formylation-driven translation of mitochondrially encoded proteins, and methylation transfer reactions in cancer cells. Second, the selective advantage provided by the (SHMT2) mitochondrial 1C metabolism versus the (SHMT1) cytosolic pathway is based on the ability of the SHMT2-dependent rate of mitochondrial serine-to-formate catabolism to regulate a master metabolic switch in nucleotide and AMPK-sensed energy metabolism. Third, the inactive apo-SHMT2 dimer is recognized as an unexpected driver of immune signaling capable of promoting inflammatory cytokine signaling via interaction with the deubiquitylating BRCC36 isopeptidase complex (BRISC). The ability of metformin to directly target the enzymatic activity of SHMT2 may improve our understanding of the SGOC metabolism-related anti-tumor actions of metformin, which should not exclusively be viewed as a downstream consequence of mCI targeting, but also as a causal upstream inhibitory effect against SHMT2-driven mitochondrial serine catabolism. Although weak, the PLP-competitive behavior of metformin hindering the formation of active holo-SHMT2 tetramers might synergize with unfavorable catalytic conditions characteristic of the folate trap imposed by vitamin B12 deficiency (e.g., decreased formation of the SHMT2 substrate THF and augmented levels of folate-trapping intermediates that inhibit SHMT2 activity) occurring upon metformin treatment.

## Data Availability

The data that support the findings of this study are available from the corresponding authors, upon reasonable request.

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
