# Peer review of "Metformin Is a Pyridoxal-5′-phosphate (PLP)-Competitive Inhibitor of SHMT2"

_cancers, 2021, doi:10.3390/cancers13164009_

Round 1

Reviewer 1 Report

I like the paper a lot. Technically well conducted.

Although there are few additional experimental approaches I would like to suggest to be test in order to claim serine/glycine driven cancer dependency can be targeted using metformin. For this I would suggest the following experimental approach. Using WT and PHGDH amplified BRCA cancer models treated with dosage ranges of metformin to show selectivity of the compound rather than using KO models. Furthermore, in these models I would like to see deuterated serine tracing to really pinpoint SHMT inhibition selectivity on dTTP generation and 13C glucose tracing to monitor inhibitory effects on de novo serine/glycine production. In addition, I would like to see the selectivity of metformin in comparison to recently identified SHMT1/2 inhibitors like SHIN1/2 inhibitors and sertraline.

Reviewer 2 Report

In the present study authors demonstrate that metformin inhibits mitochondrial SHMT2 activity by competing with pyridoxal 5'-phosphate binding and transition from dimeric to tetrameric form. The results were obtained using recombinant enzyme, cancer cell and molecular docking and dynamics simulation. The topic and the results are of interest. The study suggest novel anticancer mechanism of metformin and possibly opens perspectives of synthesizing related biguanide derivatives with more potent SHMT2-inhibiting properties. However, there are also some concerns to be addressed.

1) Tests used for statistical analysis should be described.

2) Concentrations at which metformin effectively inhibited the enzyme were relatively high and are unlikely to be obtained in vivo in metformin-treated patients. Potential therapeutic/clinical implications of the findings should be discussed.

3) There are many PLP-dependent enzymes. Did the authors performed any molecular docking/dynamic simulation about the possible effect of metformin on their activities?

4) It would be reasonable to examine if metformin has any antiproliferative effect in these cell lines correlated with SHMT2 inhibition.

Round 2

Reviewer 1 Report

Fulfilled all the reviewers wishes, and think the paper did improve well mechanistically!